# Metabolic Profiling of CHO Cells during the Production of Biotherapeutics

**DOI:** 10.3390/cells11121929

**Published:** 2022-06-15

**Authors:** Mathilde Coulet, Oliver Kepp, Guido Kroemer, Stéphane Basmaciogullari

**Affiliations:** 1Sanofi R&D, 94400 Vitry-sur-Seine, France; mathilde.coulet@sanofi.com; 2Metabolomics and Cell Biology Platforms, Gustave Roussy Cancer Center, 94800 Villejuif, France; captain.olsen@gmail.com; 3Institut Universitaire de France, Centre de Recherche des Cordeliers, Equipe Labellisée par la Ligue Contre le Cancer, Université de Paris Cité, Sorbonne Université, Inserm U1138, 75006 Paris, France; 4Department of Biology, Institut du Cancer Paris CARPEM, Hôpital Européen Georges Pompidou, AP-HP, 75015 Paris, France

**Keywords:** immunotherapy, monoclonal antibodies, metabolomics, industrial production, process optimization

## Abstract

As indicated by an ever-increasing number of FDA approvals, biotherapeutics constitute powerful tools for the treatment of various diseases, with monoclonal antibodies (mAbs) accounting for more than 50% of newly approved drugs between 2014 and 2018 (Walsh, 2018). The pharmaceutical industry has made great progress in developing reliable and efficient bioproduction processes to meet the demand for recombinant mAbs. Mammalian cell lines are preferred for the production of functional, complex recombinant proteins including mAbs, with Chinese hamster ovary (CHO) cells being used in most instances. Despite significant advances in cell growth control for biologics manufacturing, cellular responses to environmental changes need to be understood in order to further improve productivity. Metabolomics offers a promising approach for developing suitable strategies to unlock the full potential of cellular production. This review summarizes key findings on catabolism and anabolism for each phase of cell growth (exponential growth, the stationary phase and decline) with a focus on the principal metabolic pathways (glycolysis, the pentose phosphate pathway and the tricarboxylic acid cycle) and the families of biomolecules that impact these circuities (nucleotides, amino acids, lipids and energy-rich metabolites).

## 1. Introduction

In 1986, Muromumab became the first FDA-approved mAb for the prevention of kidney transplant rejection. Since 1990, and especially since the approval of the first fully human antibody in 2004, the number of mAbs available in the pharmacopeia has increased. Biotherapeutics have been used for the treatment of a variety of medical conditions including cancers, organ transplants and autoimmune, cardiovascular, respiratory and neurological diseases, which is witnessed by the steady increase in the number of FDA-approved treatments [1]. The pharmaceutical industry has made great progress in developing reliable and efficient bioproduction processes that meet the demand for new biotherapeutics including monoclonal antibodies (mAbs). Sixty-eight new mAbs were approved between 2014 and 2018, thus representing more than 50% of all new biopharmaceutical products on the market, and mAbs represented the most lucrative product class, with a revenue of $123 billion registered in 2017 [2].

mAbs are complex structures and require proper folding, assembly and post-translational modifications, such as glycosylation, to ensure their functionality and efficacy. Unlike microbial systems that are limited in post-translational modifications, mammalian cell lines are apt to generate diverse functional recombinant glycoproteins, with Chinese Hamster Ovary (CHO) cells being used for the production of 84% of FDA-approved biotherapeutics in 2018 [2].

The culture of CHO cells in bioreactors is a tightly regulated process that has seen significant advances since the 90s [3]. The steps involved in the development of a cell culture production process include the design and the selection of a stable protein-secreting cell line, the optimization of media and culture operating conditions at a small scale using medium- to high-throughput screening methods and finally the upscaling of the process, which must comply with good manufacturing practices (GMPs) [4,5]. Despite major advances in cell culture processes and the optimization of cell culture media, the cellular response to changes in the culture environment and the subsequent impact on cell growth and productivity needs to be better understood in order to fully optimize the production of biologics.

In this regard, multidimensional ‘omics’ approaches are powerful tools used to improve our knowledge and to design suitable strategies for unlocking the full potential of CHO cells. Genomics, proteomics, transcriptomics and metabolomics are recent and complementary fields of study that have been instrumental in deciphering biological mechanisms influencing cell growth in bioreactors. Metabolomics is a promising approach in the bioproduction field, as it detects the downstream products of the other ‘omics’ sciences (genomics, transcriptomics and proteomics for the characterization of DNA, RNA and proteins/enzymes, respectively) and is believed to accurately mirror the cellular phenotype. Metabolomics involves the intracellular and extracellular quantification of small molecules called metabolites [6], concentrations of which strongly vary as a function of cellular responses to environmental changes [7]. The standardized preparation of the samples, be it debris-free culture supernatants or washed cells, is the first step of metabolomic analysis. This is followed by the metabolic profiling of the samples either by nuclear magnetic resonance (NMR) or mass spectrometry (MS). A bioinformatic analysis then connects metabolites to known metabolic pathways and quantifies metabolic fluxes [8,9]. Two distinct methodologies called untargeted and targeted metabolomics are widely used for this purpose. Untargeted metabolomics is an unbiased analysis measuring all detectable metabolites present in the sample and can enable the discovery of new molecules impacting cell metabolism. Targeted metabolomics focuses on groups of known metabolites that are chemically defined. Both methods are complementary and are often used in combination to gain maximum insights [10]. Metabolomics analysis can also be coupled with the metabolic labeling of the cell substrate by adding non-radioactive isotope tracers (usually synthetic metabolites labelled with carbon-13, ^13^C) to the culture media prior to mass spectrometry analysis. This combined approach allows for metabolic flux analysis (MFA) and the calculation of reaction rates as well as the identification of precursor–product relationships among metabolites [11].

The kinetic analysis of the metabolome of cells at different time points of the cell culture is key to the improvement of processes. This is particularly true for the detection of the appearance and/or exhaustion of potential fate-changing molecules at the switch from the exponential growth phase to the stationary phase, which is characterized by low proliferation, and the final decline of the culture, which is characterized by a high rate of cell death. Many studies managed to identify such markers [11,12,13,14,15,16,17,18,19] and this paved the way for an increase in production yield through the genetic engineering of cells or by changing the medium composition and culture parameters [20,21,22]. However, the metabolomic studies reported in the literature are heterogeneous in their design and technology (Appendix A), rendering the direct comparison of results difficult.

This review aimed to discuss the results obtained by mass spectrometry metabolomics and ^13^C MFA studies of CHO cells listed in Appendix A. Key results obtained for each phase of growth (exponential growth, stationary phase, decline) regarding catabolism and anabolism are presented. Emphasis is placed on the main metabolic pathways of mammalian cells (glycolysis, the pentose phosphate pathway (PPP) and the tricarboxylic acid (TCA) cycle) and on the families of biomolecules that impact these pathways (nucleotides, amino acids, lipids and energy-rich molecules). Key changes marking the switch from exponential growth to the stationary phase and then to decline are highlighted. The results obtained in distinct experimental settings are also compared (various cell lines, clones, media and culture strategies including batch, fed-batch and continuous) to identify common trends affecting the CHO cell metabolome during culture in bioreactors.

## 2. The Exponential Growth Phase: High Nutrient Uptake Enabling Cell Growth

The exponential phase is characterized by a steep increase in the viable cell density (VCD) in the cell culture. This growth is possible because nutrients are available in the culture medium, from which (1) catabolism generates energy and (2) various substrates are used for biomass generation. Both steps will be described in this section, and the main metabolic trends are summarized in Figure 1.

### 2.1. Carbon Source Catabolism Generates Energy for the Cells as Nutrients Are Largely Available (Step 1)

MFA studies have shown that glucose contributes to 70% of the total carbon influx, while pyruvate from medium serves as an additional carbon source at the beginning of the culture, with 80% of it being consumed in the first day of culture [18,19]. Similarly, another study concluded that glucose represents 65% of the entering carbon, with 10% coming from glutamine and 25% from other amino acids [23]. A lower contribution was also reported (40–50%) [24,25], although this can partly be explained by the fact that glucose consumption can vary simply due to different culture conditions, specifically, in the present case, culture volumes [26]. Besides these subtle differences, publications overall agree on the fact that glucose constitutes the main carbon source for growth.

Glucose taken up by cells can be phosphorylated and supplied to the glycolysis pathway for ATP production or to the pentose phosphate pathway (PPP), which contributes to redox homeostasis and biosynthesis. Numerous studies report that during the exponential phase, glucose is mainly oxidized via glycolysis, which results in the formation of pyruvate. A significant portion of the pyruvate formed is converted into lactate, which is secreted and acidifies the medium, resulting in growth inhibition, with the rest being used to supply the TCA cycle [11,16,19,23,24,26,27,28,29,30,31,32,33,34,35,36,37]. This metabolic state was previously recognized as the Warburg effect and characterizes cancer cells that consume high levels of glucose. Based on the potential energy load of glucose, it can be considered that, in the sense, the carbon flux is “wasted” by the cells, in that it is not targeted to the TCA and yields high lactate levels instead. This lowers the production of the C_4-6_ precursors necessary for biomass generation [38,39].

It has been estimated that 75% of glucose consumption goes toward lactate production in cultures of adherent and suspension-growing CHO cells [11,33,40]. In perfusion cultures, however, a more balanced proportion has been measured, with 55% of the pyruvate being estimated to generate lactate and 45% entering the TCA cycle [32]. On the contrary, MFA studies on batch/fed-batch cultures report that 25–40% of cytosolic pyruvate is converted into lactate by the action of lactate dehydrogenase, an effect that is observed for both CHO WT cells and protein-producing CHO derivatives [18,23,41], while 50–60% of the pyruvate pool is channeled into the TCA cycle. These discrepancies might be attributed to different media compositions. For instance, it has been shown that glutamine provided in the media can strongly influence glucose and pyruvate metabolism [42]. It was also shown that the overexpression of *bcl2*, an anti-apoptotic gene, can influence the proportion of pyruvate entering the TCA and, as a result, lactate formation [34]. A number of studies also analyzed the differences between high and low producer clones. Certain studies revealed that (i) high producers possess higher levels of intracellular NADH, suggesting a higher level of glycolysis in combination with the TCA cycle and/or oxidative phosphorylation [43], and (ii) some differences exist in the timing of glucose consumption and lactate formation between high and low producer cell lines, with high producers consuming lactate earlier in the culture compared to low producers [19]. Only minor differences were reported in another study, suggesting that further investigations are needed to clarify how productivity might be correlated to the early phases of glucose metabolism [44].

At the pyruvate branch point, it was estimated that about 75% of pyruvate comes from glycolysis, while 25% results from the conversion of malate by the malic enzyme and a minor fraction from the degradation of amino acids such as serine or cysteine [40,41,42]. A total of 90% of pyruvate flux would enter in the TCA following its conversion to acetyl-coA, with the remaining 10% entering the TCA cycle after conversion into oxaloacetate by pyruvate carboxylase [32,33,42].

At the late exponential phase, a decrease by one-third in terms of glucose consumption and glycolytic activity was measured in favor of lactate uptake [18,37]. Interestingly, it was possible to prolong the exponential phase by the adding pyruvate and amino acids to the medium [16].

The pentose phosphate pathway (PPP) represents an alternative route of glucose oxidation that regenerates the NADPH that contributes to the redox balance in the cytoplasm and C5 sugars that are involved in biosynthetic reactions, in particular nucleotide biosynthesis. Pentose phosphate pathway (PPP) activity has been characterized during the exponential phase, and would appear that its contribution to the overall carbon flux depends on the experimental models. In the perfusion mode, the proportion of glucose entering the PPP was estimated at 20–40% [32], while studies conducted in batch/fed-batch conditions reported a very low or negligible contribution from glucose to this pathway [11,33,34,40]. This is in contradiction with one study conducted on fed-batch cultures, which reported that 80% of glucose was engaged in the PPP pathway, accounting for most of the cytosolic NADPH generation during this growth phase [23]. Thus, these studies reported major differences in terms of glucose utilization between glycolysis and PPP, perhaps reflecting variations in the composition of media, including in terms of glutamine levels [42], medium supply in fed-batch cultures, specific clones or yields in the production of recombinant proteins.

Glycolysis oxidative reactions fed by glucose or PPP intermediate species result in the generation of pyruvate that can enter the TCA cycle as acetyl-CoA. As mentioned above, TCA cycle fluxes are low during the exponential phase, with a large portion of pyruvate being used for lactate production [11,13,17,34]. As a possibility, any overabundant glucose available during initial cell culture might be used for energy maintenance through lactate production rather than for the TCA cycle and oxidative phosphorylation. It was also hypothesized that the TCA cycle might be impaired in transformed CHO cell lines due to a reduced capacity to convert citrate into α-ketoglutarate (AKG) [16,45]. Another bottleneck might exist between malate and oxaloacetate due to a limited enzyme capacity, which would result in limited malate-aspartate shuttling and, hence, NAD regeneration in the cytosol. In addition, this limitation in oxaloacetate supply prevents the incorporation of mitochondrial pyruvate into the TCA cycle, resulting in the conversion of pyruvate conversion into lactate in the cytoplasm [44]. Although lactate production is significant, as mentioned earlier, the TCA is active, with the main carbon source coming from glucose oxidation to pyruvate and the rest from glutamine and essential amino acids [23]. Interestingly, it has been shown that the secretion of TCA intermediates (succinate, malate, citrate and fumarate) represents approximately the same level of TCA cycle flux during the exponential growth and stationary phases, possibly indicating that the TCA cycle is operating at close to maximal enzymatic capacity throughout cell culture [44]. However, this conclusion was reached upon using ^1^H-NMR, which is comparatively less sensitive than MS. It has also been suggested that the culture scale impacts the metabolism of cells. Cells growing in production-scale bioreactors may rely more on glycolysis for energy production than cells growing at lab scale that synthesize more TCA cycle intermediates [46].

Regarding nucleotides and nucleosides at the exponential phase, little information is available in the literature. Up to 90% of ATP production has been suggested to result from the TCA cycle [41], with a downward trend along the exponential phase [17]. While ADP and AMP concentrations were found to be similar in the cytosol and mitochondria, ATP concentration was higher in the cytosol than in the mitochondria, suggesting that the transport of ATP out of the mitochondria might render ATP formation thermodynamically efficient [30]. At this phase of growth, when comparing high and low producer clones, steady-state ATP concentrations are comparable, but the production and consumption rates correlate with productivity in *Escherichia coli* and CHO cells [43]. The addition of adenosine monophosphate (AMP) or guanosine monophosphate (GMP) to the culture media led to a 3-fold induction of caspase activity, the highest observed for all metabolites tested, contrasting with the addition of cytidine monophosphate (CMP) and uridine monophosphate (UMP), which had no effects [47]. This pro-apoptotic effect was also reported in other studies on CHO cells [48] and IEC-6 intestinal epithelial cells [49]. This means that, even in the presence of nutrients and in the absence of high levels of ammonia and lactate, toxic metabolites can cause cell death.

The consumption and production of amino acids has been studied in CHO cells during the exponential growth phase (Table 1), and amino acid-relevant metabolic fluxes were described to be significantly lower than glycolytic and TCA cycle fluxes [32]. MFA studies reported consistent percentages in terms of the contributions of amino acid to TCA cycle replenishment, with glutamine being the main carbon source, and it is known that its availability in the culture media, together with asparagine or serine, has a high impact on the metabolism of CHO cells [25,50,51]. Glutamine was found to contribute to about 40% of the total carbon supply of the TCA cycle. The mitochondrial glutamate pool originates from both the transport of cytosolic glutamate into the mitochondria and from the transport of glutamine into the mitochondria followed by hydrolysis to glutamate. Glutamate is converted into α-ketoglutarate by the mitochondrial glutamate dehydrogenase, which is fed into the TCA cycle [23]. All the other amino acids contribute about 10% and enter the TCA cycle at either the acetyl-CoA, α-ketoglutarate or oxaloacetate branch point. Very similar amino acid catabolic contributions were reported in induced and non-induced cells [41]. For low and high producer clones, consistent results were also observed, with asparagine and glutamine generating about 30% of the citrate pool and 50% of malate and succinate labelling, suggesting their major role in ATP generation and the replenishment of the TCA cycle at the exponential phase of growth but not biomass production [19]. The total contribution of amino acids together with glucose to incoming carbon flux is thought to remain constant over the entire culture (between 30% and 50% of the total carbon) with the uptake of other amino acids increasing after glutamine is depleted [18]. Another model estimated that 35% of the central carbon metabolism is fed by amino acid catabolism, with glutamine representing 10% [23]. More precisely, 60% of the aspartate formed was transaminated to yield oxaloacetate during exponential growth, decreasing to 50% in the stationary phase. Alanine was produced for nitrogen detoxification in the exponential phase, corresponding to 6% of pyruvate usage.

On average, 25% of the essential amino acids consumed were not supplied for biomass or IgG production during exponential growth and instead generated TCA cycle intermediates [44]. This interpretation is supported by a second study in which the rate of glutamine uptake during the early exponential phase greatly exceeded the biosynthetic demand for biomass or antibody production, which reflects its use in catabolic energy production [18].

A study comparing cultures fed with high or low concentrations of glutamine showed the importance of this metabolite, as major differences in pathway fluxes were observed. These variations impacted strongly on the specific productivity, which was 2-fold higher for cultures fed with low concentrations of glutamine, and showed how sensitive CHO cell metabolism is to glutamine levels [42].

The replenishment of the TCA by amino acids generates intermediate molecular species. At the end of the exponential phase, when cell density is high and lactate concentration is low, these intermediates tend to accumulate, and their inhibitory effect on cell growth has been demonstrated, with indole 3-carboxylate and isovalerate being the most potent (refer to Table 1, Table 2 and Table 3 for more details) [21].

In addition to their role in the replenishment of the TCA and protein biosynthesis, amino acids serve as glutathione precursors. Their metabolism thus has a major impact on cell protection against oxidative stress caused by ROS [54]. In this respect, glutathione is one of the key metabolites to consider, as the equilibrium between its oxidized (GSSG) and reduced (GSH) forms influences cell growth. Both forms are present at high concentrations during the early exponential phase at a 1:5 ratio. The GSSG/GSH ratio is then balanced during the exponential phase [31,55]. Consistently, several publications reported the accumulation of GSSG in media during the exponential growth phase, which then continued during the stationary phase. GSSG is known to be a marker of oxidative stress and to induce apoptosis, presumably explaining its correlation with caspase activity [12,17,20,47]. An intracellular decrease in GSSG and GSH concentrations has also been reported [17,18]. This is consistent with the accumulation of GSSG observed in media as it is excreted by cells, and extracellular GSSG might then account for cell death in prolonged cultures [38]. Additionally, it has been suggested that the decrease in GSH concentration observed during the late exponential phase together with the decrease in NADPH concentration triggers an increase in PPP activity before the peak of TCA cycle fluxes. In favor of this hypothesis, a close correlation between PPP and TCA activities was observed during the culture [18]. Based on a study focused on the redox status in the cell, it has also been suggested that cell ageing is not responsible for the increase in glutathione metabolism and ROS management, but that high cell densities are [54].

### 2.2. The Use of Energy Enables Biomass Production (Step 2)

To our knowledge, no metabolomics studies focused on nucleic acid and protein synthesis. This section will focus on lipid metabolism, which is extensively covered by metabolomics studies.

One group of studies measured an increase in intracellular concentrations of medium and long chain fatty acids in line with the need for cell growth during the exponential phase. In line with this observation, the contribution of fatty acid oxidation to the acetyl-CoA mitochondrial pool was found to be negligible during the exponential phase [11], with acetyl-CoA being mainly directed toward fatty acid synthesis at this phase of cell growth [23]. A large fraction of citrate was found to be transported to the cytosol for its conversion into oxaloacetate and acetyl-CoA [31,33]. An MFA study calculated that about 25% of the citrate synthase flux is channeled via citrate lyase to the biosynthesis of fatty acids. In this study, the TCA cycle intermediates mostly fueled lipid synthesis and rendered negligible amounts of amino acids [32]. A study performed on adherent CHO cells demonstrated that acetyl-CoA produced from glutamine does not contribute significantly to fatty acid biosynthesis, suggesting that other sources of carbon such as glucose do [33]. As a component of the glycerol-based phospholipids that compose cell membranes, glycerol was also found to accumulate extracellularly, representing 5% of the carbon engaged in glycolysis [26,44]. In their review, Pereira et al. (2018) stated that glycerol released into the medium results in a loss of carbon to the cells. For the authors, glycerol can be regarded as a storage compound or redox sink in the regeneration of NAD pools [38,39]. Other studies have shown that choline, choline phosphate, ethanolamine phosphate, cytidine-diphosphate-choline (CDP) and CDP-ethanolamine are almost depleted during the exponential phase, while other precursors of the cell membrane constituents are accumulated [12,17]. It was suggested by Selvarasu et al. that this might be a reflection of poor membrane lipid metabolism regulation in CHO cells and that this might trigger the transition from the exponential to the stationary phase [17]. In support of this hypothesis, an increase in extracellular phosphocholine and glycerol-3-phosphocholine was reported and correlated with intracellular caspase activity and apoptosis [44,47]. A thorough investigation of the altered metabolic flux, including in the glycerophospholipid metabolic pathway, might help in the design of strategies to enhance cell culture viability.

Unlike the exponential growth phase, for which some carbon flux consensus can be drawn across cell culture models, the stationary and decline phases are characterized by metabolic changes that appear to be cell culture model-dependent. Readers are encouraged to refer to Appendix A and compare their results to those obtained in relevant cell culture models.

## 3. The Stationary Phase: Stabilized Growth and High Recombinant Protein Production

### 3.1. Alternative Carbon Source Catabolism Enables Energy Production and Cell Maintenance

As a general trend, most of the carbon and nitrogen sources were found to be consumed at lower rates during the stationary phase compared to the exponential phase [34,40,44]. Thus, a 5-fold decrease in glycolytic activity was reported to be accompanied by an equivalent reduction in glutaminolysis and AKG production [33]. The stationary phase also witnesses a major shift in the metabolism of lactate, which is not so much produced but consumed, enabling TCA cycle replenishment, as the glucose concentration decreases [11,16,17,18,31,33,34,44,56]. Glucose consumption continues, even though at a decreased rate, until it is fully depleted [11,27,41,44]. One report, namely [18], indicated a slight decrease in glutaminolysis, while glucose fluxes do not drop below 50% of their initial rate. A decrease in glucose consumption and glycolysis fluxes by one-third was observed at the late exponential phase and the stationary phase [18]. These key characteristics were also observed for antibody-producing GS-NS0 mouse myeloma cells. In such cells, lactate was consumed earlier when proliferation was artificially stopped, which also correlated with increased productivity [28]. Evidence was provided that glucose is not depleted during the stationary phase but at the entry in the decline phase [16]. The depletion of asparagine and aspartate was reported to occur at the entry into the stationary phase, well before the exhaustion of glucose. Moreover, pyruvate was identified as a growth-limiting metabolite, meaning that its depletion marks the transition from the exponential to the stationary phase [44]. The depletion of pyruvate was confirmed in a more recent study [31].

PPP fluxes increase during the stationary phase compared to the exponential phase by a factor of 5 to 6-fold, corresponding to 30% of the glucose uptake rate [11,33] or possibly even more [13,18,34]. Sengupta et al. hypothesized that during the stationary phase, cells might experience more oxidative stress, leading to a higher utilization of the PPP to generate NADPH for reductive reactions. This is supported by a study on *Escherichia coli* that showed that the bacteria fail to fight oxidative stress during the stationary phase because of the downregulation of genes involved in aerobic electron transport [57]. One recent report showed that the downregulation of the PPP, as opposed to its activation, occurred during the stationary phase in CHO cells [31], but this relationship has not been extensively studied in CHO cells and deserves further investigation.

Most studies agree that succinate, malate, citrate and fumarate accumulate during the stationary phase in the culture supernatant [12,20,27,46,58]. At this stage, carbon influx into the TCA cycle is provided by glucose (50%), glutamine (40%) and other amino acids (10%) entering the TCA cycle in the form of acetyl-CoA or anaplerotic substrates including malate, α-ketoglutarate or oxaloacetate [41]. More importantly, a switch in pyruvate utilization is observed from lactate production and secretion to its translocation into the mitochondria and use in the TCA cycle, which reaches its peak activity [13,18,34]. Such TCA cycle activation may be secondary to exacerbated oxidative stress, resulting in a decreased NADH/NAD^+^ ratio and the consequent counter regulation of the regeneration of the NADH pool.

In addition to the publications that report the increased activity of the TCA cycle in the stationary phase, equivalent or decreased activity has also been reported. For example, MFA performed on adherent CHO cells during the stationary phase led to the conclusion that most of the pyruvate is diverted to the TCA cycle (instead of yielding lactate), with no impact on the TCA cycle, which operates at similar levels as during the exponential phase [11,33]. This was also observed in a study using ^1^H-NMR, where the secretion of succinate, malate, citrate and fumarate were found to represent the same proportion of the TCA cycle flux as during the exponential phase. This indicates that in these models, the TCA is operating at close-to-maximum enzymatic capacity throughout the culture time [44]. In other studies, the activities of glutamate dehydrogenase and pyruvate carboxylase, key anaplerotic enzymes, were found to be reduced 2-fold, and the flux from α-ketoglutarate to succinyl-CoA was also lowered to 50% [11,33]. Such decreased TCA cycle activity has also been observed in other works [16,17] and led to the conclusion that glucose might be redirected to other metabolic pathways to enable ATP generation and NADPH oxidation in the stationary phase, during which proliferation is limited and productivity is increased (see below).

Regarding amino acids, observations made during the stationary phase of growth are reported in Table 3. According to a number of studies, anaplerosis might be strongly decreased during the stationary phase compared to exponential phase, resulting in 3-fold lower malic enzyme fluxes and negligible fluxes of amino acids toward the TCA cycle [11,13]. However, another more recent study estimated that the consumption of essential amino acids for anaplerosis increases from 25% during the exponential phase to 35% during the stationary phase, pointing to the key role of amino acids in energy production throughout the culture [44]. These conflicting results demonstrate that cell metabolism across culture phases might be model and clone dependent.

### 3.2. The Use of Energy Enables Recombinant Protein Production

The production of recombinant proteins by CHO cells ramps up during the stationary phase, accounting for 15% of incoming carbon fluxes, in contrast to only 3% during the early exponential phase [18].

In contrast to the exponential growth phase, about 25% of pyruvate formation results from lactate consumption during the stationary phase, with other minor sources being malate and serine together with pyruvate uptake from the media [33,40,44]. The entry of pyruvate into the TCA cycle by conversion into acetyl-CoA was measured to be similar in the exponential and stationary phases. On the contrary, its conversion into oxaloacetate via pyruvate carboxylase activity was found to be undetectable during the stationary phase [33].

Regarding glycerophospholipids, one study reported that the transition between exponential growth and the stationary phase is marked by the intracellular appearance and then accumulation in media of glycerol-3-phosphate and glycerol, the latter being presumably synthesized from glycerol-3-phosphate [14,16]. Carinhas et al. estimated that 25% of the carbon engaged in glycolysis ends up in glycerol [44]. The surge in glycerol concentration observable at the transition from the exponential to the stationary phase was interpreted to reflect the lipid biosynthesis that is required for cell proliferation-associated membrane formation, the turnover of membrane lipids as well as the secretion vesicles used for protein excretion [14]. An increase of glycerophosphocholine, a product of the degradation of phosphatidylcholine, which is a major component of the plasma membrane, was also observed. This was interpreted to reflect cell membrane degradation and cell growth limitation [17]. Surprisingly, fatty acid biosynthesis was found to be as high during the stationary phase as during the exponential phase, suggesting another role in addition to cell growth that was not elucidated by the authors [33]. This is consistent with the hypothesis that lipid synthesis might reflect the need for membrane lipid turnover (plasma membrane and intracellular membranes involved in vesicular trafficking).

## 4. The Decline Phase: Media Exhaustion Resulting in Cell Death

Relatively few studies have investigated CHO metabolism during the decline of cell cultures. The final phase was found to be accompanied by the exhaustion of glucose, with a consequent decrease in glycolytic intermediates [16]. Pyruvate is depleted, while lactate remains relatively constant in batch culture [14]. In fed-batch [14], and HiPDOG (hi-end pH delivery of glucose) culture conditions [18], lactate utilization was observed during the decline phase. In this latter study, PPP flux was maintained even when cell density began to decline.

Regarding the TCA, Matuszczyk et al. have demonstrated the depletion of cytosolic pyruvate while mitochondrial pyruvate was available at higher concentrations, suggesting high TCA cycle activity [30]. This is in line with another study reporting that TCA cycle activity is even higher during the decline phase than during the early exponential phase [18]. On the contrary, a significant decrease in the metabolites of the TCA cycle compared to the other culture phases was reported in one study, and this was interpreted to be due to less carbon skeleton from glucose fueling the TCA during this phase [16]. This discrepancy may be explained by the fact that the former study was focused on an early phase of decline while the latter was focused on a terminal phase of decline and medium exhaustion.

Regarding nucleotide metabolism, ATP, ADP and AMP concentrations decreased with the culture time until the decline phase, but their distributions across the cytoplasmic and mitochondrial compartments differed. The ATP concentration was found to be higher in the cytosol than in the mitochondria during this culture phase, unlike in the cases of ADP and AMP, which were found at similar concentrations in both compartments [30].

Regarding amino acids, observations made during the decline phase are reported in Table 3 There is no comprehensive analysis of fatty acid metabolism during the decline phase, but a peak of medium and long chain fatty acid concentrations was reported at this phase [31]. Finally, regarding the oxidative state of the cell, an increase in oxidized glutathione was observed, leading to an unfavorable GSSG/GSH ratio for ROS management [31].

In addition to a comprehensive understanding of cell metabolism during the decline phase, which needs further investigation, several strategies have been suggested to extend the productive phase of the cell suspension. For example, one study recommended the growing of cells in galactose-containing medium, as it is metabolized when glucose is depleted, which enables the maintenance of cell viability [59]. Another study suggested the addition of an anti-apoptotic agent to avoid excessive cell death [46].

## 5. Conclusions

The comparison of all of the metabolomic studies on CHO cells reveals general trends for each of the phases of cell cultures, as they highlight the most important shifts from expansion to stabilization and then to final viable cell density decline. Overall, the exponential phase of growth corresponds to a state of high consumption in terms of nutrients, in particular glucose and glutamine. This enables high anabolic activity, a balanced redox state and the formation of excess energy that can be utilized for biomass production, favoring intense cell proliferation. The oxidation of glucose via glycolysis is high, enabling bioenergetic homeostasis and energy “storage” by means of lactate production, contrasting with the moderate pyruvate-mediated fueling of the TCA cycle, which is instead replenished by anaplerosis due to the availability of all amino acids. A characteristic of the growth phase is low mAb production, which makes the subsequent stationary phase the most interesting with regard to the synthesis of recombinant proteins. This reduced proliferation enables a higher investment in cellular energy for biosynthesis. The stationary phase is characterized by a comparably low consumption of carbon and nitrogen sources and reduced glycolytic activity. In contrast, TCA cycle fluxes are increased during this phase, with lactate consumption supplying the acetyl-CoA pool. In this phase, the role of amino acids in TCA replenishment deserves further investigation. Also, compared to the exponential phase, PPP is largely activated and thus generates reducing equivalents such as NADPH and glutathione, which fight oxidative stress during the stationary phase. The final decline of cultures is marked by the exhaustion of many metabolites and the accumulation of toxic products.

Importantly, this review also highlights important discrepancies between results from various studies. These discrepancies may be explained by the unique features of the CHO cell lines derived from a single host, acquired by genetic drift in separate laboratories [60]. Differences in terms of the media used and the culture strategies have also shown to give rise to genetic and epigenetic diversity, and this could explain phenotypic differences when distinct processes are used for the production from the same clone [61,62]. This diversity is known to have a major impact on the metabolomic profile of CHO culture.

For instance, principal component analysis performed on cell culture supernatants of six mAbs producing cell lines derived from two distinct hosts highlighted that variations in metabolic profiles were highly dependent on the cell line’s host lineage [22]. When comparing the metabolic profile of high and low producer clones, key differences were also identified [19,43]. In the first study, Chong et al. [43] identified seven metabolites involved in the main metabolic pathways of CHO cells (glycolysis, the PPP and the TCA cycle), associated with high productivity. It was suggested that the abundance of these metabolites improved the control of the oxidative state, thus increasing the production of recombinant mAb. Furthermore, Dean et al. [19] highlighted several metabolic differences between high and low producers’ cell lines. The low producer clone consumed more glucose at the start of the culture, while accumulating less lactate. In addition, more than 50% of the intracellular lactate originated from extracellular pyruvate in the high producer clone, whereas it represented only 30% in the low producer clone. After four days of culture, the high producer was able to consume the accumulated lactate, which was not the case for the low producer. More importantly, TCA metabolites were replenished from glucose in high producer clones, suggesting increased activity in terms of the TCA cycle, likely representing a hallmark of high production [19,63]. It was consistently shown that the production of the recombinant protein correlated with a more efficient utilization of glucose for the TCA cycle [41].

Specific gene silencing or over-expression was also shown to alter cell metabolism. For example, the endogenous expression level of the bcl-2 anti-apoptotic gene [34] or the overexpression of malate dehydrogenase II [20] resulted in diminished lactate accumulation and decreased NADH abundancy when compared with control cells.

Finally, the diversity in terms of media formulation across the studies referenced in this review might explain some of the discrepancies identified. For instance, the peak lactate concentration can be decreased by the use of a chemically defined CHO medium when compared to a chemically defined hybridoma medium [64]. It was also shown that maintaining glucose concentrations to almost negligible levels in HiPDOG cultures enabled the maintenance of lactate, osmolarity and ammonia below inhibitory levels [21]. Similarly, the concentration of glutamine, another major carbon source for cells, in the culture medium was shown to have a major impact on the CHO metabolic states [42]. In low glutamine-containing medium, glycolytic fluxes were significantly increased while PPP was decreased 2-fold when compared to cultures growing in high glutamine medium. Pyruvate usage by the TCA cycle instead of lactate production was more important in low glutamine medium cultures, which was also observed by Kirsch et al. [25]. An increase in TCA flux in such culture conditions was also reported in other studies, including in CHO-DHFR cells [50].

Nevertheless, the comparison of results obtained by MS metabolomics and ^13^C MFA to results obtained using other methods can be an efficient strategy when used to improve our knowledge of CHO cells metabolism and identify main trends. Future investigation must address the cause–effect relationships between metabolic shifts and mAb production yields. This could be achieved by connecting information obtained on experiments using several “omics” tools beyond metabolomics such as genomics, epigenomics, transcriptomics and proteomics [65]. Moreover, the increasing availability of single-cell “omics” may increase our knowledge of individual cells, thus allowing for a refined analysis of likely heterogeneous cell populations, eventually improving clone selection.

## Figures and Tables

**Figure 1 cells-11-01929-f001:**
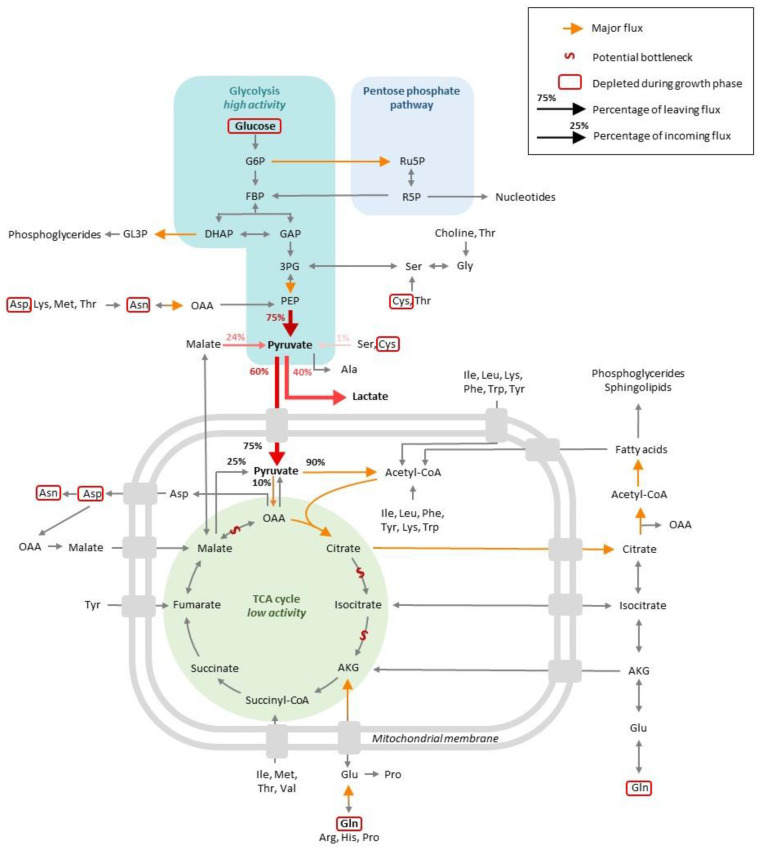
Global view of the central CHO cell metabolism at the exponential phase of growth.

**Table 1 cells-11-01929-t001:** Amino acids and amino acid derivatives at the exponential phase.

Amino Acid Derivative	Behavior	Reference	Comments
Alanine	Intracellular production and accumulation in media	[11,14,16,17,18,32,36,37,41,42,44]	
	Production mainly from cytosolic pyruvate	[23]	
Arginine	Intracellular consumption leading to concentration decrease in media	[12,20,32,41]	Different conclusion might be due to feeding method and medium composition that is specific to [17]
	Accumulation from late-exponential phase onwards, indicating over-supply in the fed-batch	[17]	
Asparagine	Intracellular consumption leading to concentration decrease in media	[17,23,25,32,36,37,41]	
	The most consumed amino acid from media. Intracellular deamination to aspartic acid generates ammonia	[44]	
	Consumption (measures performed on cell substrate). Represents 5% of incoming carbon source. Linked to aspartic acid production	[18]	
	Intracellular concentration increase in early exponential phase In terms of uptake from the media, the highest compared to other growth phases. In high producer clone, consumption beyond stoichiometric requirements together with glutamine to replenish the TCA intermediates	[19,31]	
Aspartic acid	Production linked to asparagine uptake (measures performed on cell substrate)	[18,31,41,42]	
	Consumption (measures performed on cell culture supernatant)	[32]	
	Consumption (measures performed on cell substrate)	[23,44]	
Cysteine	Consumption (measures performed on cell culture supernatant)	[41]	
Glutamine	Cells use more glutamine when cultivated in a media containing more glutamine	[25,42,52]	
	Consumption during all culture phases, with the highest during the exponential phase; source of lactate formation	[11,19,32,36,37]	
	Main amino acid consumed	[23,41]	
	Consumption linked to glutamic acid production	[18]	
	Uptake from the media, the highest compared to other growth phases. In high producer clone, consumption beyond stoichiometric requirements together with asparagine to replenish the TCA intermediates	[19]	
Glutamic acid	Intracellular production leading to increased concentration in media	[11,23,31,32,36,41,42]	
	Production associated with glutamine uptake	[18,40]	
Glycine	Accumulation in media during culture	[14,16,17,23,41,42]	The low uptake identified in [32] might be due to the analysis method (NMR) used, which is different from that used in other studies focused on glycine metabolism
	Accumulation in media during culture; produced by serine catabolism	[44]	
	Low uptake from media	[32]	
	Its accumulation in media together with the other identified metabolites leads to growth inhibition in HIPDOG culture	[21]	
Histidine	Uptake from media	[32,41]	
Isoleucine	Uptake from media	[23,32,41]	
Leucine	Uptake from media	[23,32,41]	The particularity of study [21] is the use of the specific HiDPOG culture process that might explain the difference observed
	Its accumulation in media together with the other identified metabolites leads to growth inhibition in HIPDOG culture	[21]	
Lysine	Uptake from media	[23,32,41]	
Methionine	Uptake from media	[23,32,41]	The particularity of study [21] is the use of a specific HiDPOG culture process that might explain the difference observed
	Its accumulation together with the other identified metabolites leads to growth inhibition in HIPDOG culture	[21]	
Phenylalanine	Uptake from media	[12,20,23,32,41]	The particularity of study [21] is the use of a specific HiDPOG culture process that might explain the difference observed
	Its accumulation in media together with the other identified metabolites leads to growth inhibition in HIPDOG culture	[21]	
Proline	Accumulation in media	[32]	In study [41], a medium different from that of the two other studies was used, potentially explaining this discrepancy
	Slight increase in concentration in media with time	[11]	
	Uptake from media	[41]	
Serine	An increase in the intracellular concentration during the early exponential phase	[31]	The particularity of study [21] is the use of a specific HiDPOG culture process that might explain the difference observed
	Uptake from media	[23,32,41,44]	
	Its accumulation together with the other identified metabolites leads to growth inhibition in HIPDOG culture	[21]	
Threonine	Uptake from media	[23,32,41]	
Tryptophan	Uptake from media	[32]	The particularity of study [21] is the use of a specific HiDPOG culture process that might explain the difference observed
	Depleted in media despite constant addition of feed	[12,20]	
	Its accumulation together with the other identified metabolites leads to growth inhibition in HIPDOG culture	[21]	
Tyrosine	Uptake from media	[23,32,41]	The particularity of study [21] is the use of a specific HiDPOG culture process that might explain the difference observed
	Its accumulation together with the other identified metabolites leads to growth inhibition in HIPDOG culture	[21]	
Valine	Uptake from media	[23,32,41]	
**Amino acid derivative**	**Behavior**	**Reference**	**Comments**
Aspartylphenylalanine	Accumulation in media; known to be toxic	[12,20]	
Glutamylalanine	Accumulation in media	[12,20]	
Glutamylphenylalanine	Accumulation in media; known to be detrimental to cell growth	[12,20]	
Formylmethionine	Accumulating in media	[12,20]	
5-L-glutamyl-L-alanine	Accumulation in media	[17]	
Dimethyl-L-arginine	Accumulation in media; associated with apoptosis	[17,53]	
N-acetyl-L-Leucine	Accumulation in media	[17]	
N-acetyl-L-phenylalanine	Accumulation in media; associated with apoptosis	[17]	
N-acetylmethionine	Accumulation in media	[17]	
N-formyl-L-methionine	Accumulation in media	[17]	
L-homocysteine	Accumulation in media. Metabolic source is methionine. Its accumulation together with the other identified metabolites leads to growth inhibition in HIPDOG culture	[21]	
3-(4-Hydroxyphenyl) lactate	Accumulation in media. Metabolic sources are phenylalanine and tyrosine. Its accumulation together with the other identified metabolites leads to growth inhibition in HiPDOG culture	[21]	
Phenyllactate	Accumulation in media. Metabolic source is phenylalanine. Its accumulation together with the other identified metabolites leads to growth inhibition in HiPDOG culture	[21]	
Indole 3-lactate, indole-3-carboxylate, 2-hydroxyburtyrate, and 4-hydroxyphenylpyruvate	Accumulation in media. Metabolic source is tryptophan. Its accumulation together with the other identified metabolites leads to growth inhibition in HiPDOG culture	[21]	
Isovalerate	Accumulation in media. Metabolic source is leucine. Its accumulation together with the other identified metabolites leads to growth inhibition in HiPDOG culture	[21,44]	
Formate	Accumulation in media. Metabolic sources are serine, threonine and glycine. Its accumulation together with the other identified metabolites leads to growth inhibition in HiPDOG culture	[21,44]	
Isobutyrate	Accumulation in media	[44]	
Ammonia	Secreted in media during exponential and transition phase, with this being correlated with glutamine and asparagine consumption. Accumulation is known to affect productivity and inhibit cell growth	[17,44]	
Ammonium	Intracellular and media accumulation mainly due to breakdown of glutamine and several amino acids; detrimental effects on growth presumably due to apoptosis induction	[11,31,38]	

**Table 2 cells-11-01929-t002:** Amino acids and amino acid derivatives at the stationary phase.

Amino Acid	Behavior	Reference	Comments
Alanine	Accumulation in media	[16]	The switch observed in [44] might be due to the analysis method (NMR), which is different to those used in other studies
	Constant increase in concentration with time	[11]	
	Switch from alanine secretion to uptake in media during stationary phase at the same time as aspartate and asparagine exhaustion	[44]	
Arginine	Intracellular level decreases during the stationary phase	[31]	
Asparagine	Depleted at entry into stationary phase. Exponential growth and antibody production continue if it is added again in the media	[14,16,36,40,44]	The asparagine concentrations used in these studies might be different, leading to its exhaustion at different times of the culture
	Intracellular consumption. Represents 8% of incoming carbon source	[18]	
	Uptake from the media, with this being lower than at exponential phase	[19]	
Aspartic acid	Below detection level once cells enter stationary phase. Exponential growth and antibody production resumes when added to media	[14,16]	The aspartic acid concentrations used in these studies might be different, leading to its exhaustion at different times of the culture
	Depleted in media during stationary phase	[44]	
Cysteine	Depletion at entry into stationary phase	[14]	
Glutamine	Consumption	[11,31]	The glutamine concentrations used in these studies might be different, leading to its exhaustion at different times of the culture
	Depleted in media at entry in stationary phase	[23,36,40]	
	Uptake from the media, with this being lower than at exponential phase	[19]	
Glutamic acid	Below detection level once cells enter stationary phase. Exponential growth and antibody production resumes when added to media	[16]	The glutamic acid concentrations used in these studies might be different, leading to its exhaustion at different times of the culture
	Constant detection	[11]	
Glycine	NA		
Histidine	NA		
Isoleucine	NA		
Leucine	NA		
Lysine	NA		
Methionine	NA		
Phenylalanine	Intracellular level dropped during stationary phase	[31]	
Proline	Slight increase in concentration with time	[11]	
Serine	Consumption from media	[44]	
Threonine	Intracellular level decreases during stationary phase	[31]	The particularity of study [21] is the use of a specific HiDPOG culture process that might explain the difference observed
	Accumulation in media identified as growth inhibitor in HiPDOG culture	[21]	
Tryptophan	Intracellular level decreases during stationary phase	[31]	
	Increased availability in media correlates with diminished viable cell density and accumulation of an intermediate during tryptophan metabolism, 5-hydroxyindolacetaldehyde (5-HIAAld), which is suspected to be an inhibitor of cell growth	[22]	
Tyrosine	Depletion at entry into stationary phase	[14]	
Valine	Intracellular level decreases in the stationary phase	[31]	
**Amino acid derivative**	**Behavior**	**Reference**	**Comments**
Acetylphenylalanine	Accumulation in media at the beginning of stationary phase. Known to be detrimental to cell growth	[12,20]	
Dimethylarginine	Accumulation in media at the beginning of stationary phase. By-product of protein degradation and can induce apoptosis	[12,20]	
N-formimino-L-glutamate	Accumulation in media as culture enters stationary phase. Metabolite of the degradation of histidine or glutamate	[22]	
Ammonia	Accumulation in media	[44]	
Ammonium	Intracellular and media accumulation mainly due to breakdown of glutamine and several amino acids; has detrimental effects on growth presumably due to apoptosis induction	[11,31]	

**Table 3 cells-11-01929-t003:** Amino acids and amino acids derivatives at the decline phase.

Amino Acid	Behavior	Reference	Comments
Alanine	Accumulation in extracellular media after addition of anti-apoptotic agent	[46]	
Arginine	NA		
Asparagine and aspartic acid	Depleted from extracellular media after addition of anti-apoptotic agent	[46]	The asparagine and aspartic acid concentrations used in these studies might be different, leading to their exhaustion at different times of the culture
	Intracellular concentration decreases compared to other growth phases	[31]	
	Depletion during stationary phase	[44]	
Cysteine	NA		
Glutamine	NA		
Glutamic acid	Accumulation in extracellular media after addition of anti-apoptotic agent	[46]	The glutamic acid concentration used in these studies might be different, leading to its exhaustion at different times of the culture
	Decreased intracellular concentration compared to other growth phases	[31]	
Glycine	NA		
Histidine and isoleucine	Extracellular concentration decreases after addition of anti-apoptotic agent	[46]	
Leucine	Exhaustion at entry into decline phase	[16]	
Lysine	Extracellular concentration decreases after addition of anti-apoptotic agent.	[46]	The glutamic acid concentration used in these studies might be different, leading to its exhaustion at different times of the culture
	Exhaustion at entry into decline phase	[16]	
	Detected; indicates over-supply	[52]	
Methionine and phenylalanine	Extracellular concentration decreases after addition of anti-apoptotic agent	[46]	
Proline	NA		
Serine	Extracellular concentration decreases after addition of anti-apoptotic agent	[46]	The glutamic acid concentration used in these studies might be different, leading to its exhaustion at different times of the culture
	Exhaustion at entry into decline phase	[16]	
	Intracellular concentration decreases compared to other growth phases	[31]	
Threonine, tryptophan, tyrosine, valine	Extracellular concentration decreases after addition of anti-apoptotic agent	[46]	
Ornithine	Detected. Known to have apoptotic properties	[52]	
**Amino acid derivative**	**Behavior**	**Reference**	**Comments**
Pyroglutamate	Extracellular concentration increases after addition of anti-apoptotic agent	[46]	
4-hydroxyproline	Extracellular concentration decreases after addition of anti-apoptotic agent	[46]	
Dimethylarginine, flutamylphenylalanine, glycerophosphocholine, hexanoglycine	Caspase activity increases after exposition of cells to these metabolites	[47]	

## Data Availability

Not applicable.

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
