# Peer review of "Metabolic Profiling of CHO Cells during the Production of Biotherapeutics"

_cells, 2022, doi:10.3390/cells11121929_

Round 1
Reviewer 1 Report
The manuscript provides a good overview of diverse publications dealing with the analysis of metabolites in culture supernatant or intracellular. This is very important work and it is also mentioned that metabolic flux analyses are a rather difficult and complex undertaking due to of different analytical methods, the use of different cultivation media, including serum-containing media, and the interpretation of at what stage of cultivation the samples were drawn (exponential, stationary, and declining phase). Finally the subspecies of the CHO host cell line plays a significant role.
A more detailed discussion of these parameters would be helpful in planning further experiments and improve the scientific quality of the paper. For example, Reinhart et al, 2019 showed that different host cell lines in 2 different media behave differently. It is also known that dhfr deficient CHO cells have other demands on the nutrient supply than CHO K1 which is demonstrated by the glutamine/glutamate cycle which is completely different. The host cell line and the medium need to be necessarily involved in Tables 2 and 3, so that a better assessment of the described discrepancies can be made.
Author Response
Dear reviewer,
Thank you for considering our review article entitled Metabolomic profiling of CHO cells during the production of biotherapeutics for publication in Cells.
Based on your comments several modifications were done on the manuscript to improve the scientific quality of this work.
- A detailed discussion of the impact of cell culture medium and cell line host lineage on the metabolomic profile of cells was included in the conclusion part. It is based on metabolomics and MFA studies that compared these parameters.
- A “Comments” column was added to Tables 1, 2 and 3. This column aims at guiding the reader for the amino acids for which different conclusions were reported. It appeared more readable to us compared to adding the host cell line and medium used for each study. This made the tables “heavy” and complicated to read. The Supplementary Table S1 is a complement to the suggestion made if a more detailed information is needed for a specific metabolite.
Reviewer 2 Report
The review by M. Coulet et al. describes key findings on the metabolism of CHO cells during different cell growth phases. As CHO cells are the most used system for producing protein drugs, understanding their metabolism is essential. The manuscript overall is well written; however, I have several suggestions:
I would transfer table 1 to supplementary material - it takes a lot of space in the manuscript and is not highly referenced in the manuscript to have it "directly under the eyes."
I think it would be easier to discuss and to understand the changes in the metabolism of amino acids and amino acid derivatives during different cell growth phases if tables 2, 3, and 4 were combined.
A schematic representation/ graphical summary of different cell growth phases, including main hallmarks in metabolism, expected phase duration, and markers of phase change, would help readers.
Other comments:
Line 116 "both steps" - what steps are kept in mind?
Line 126 "with 25% coming from amino acid and 10% from glutamine" - glutamine is also amino acid; maybe rephrasing would help?
The "Figure 2B" referenced in the text (line 323) is not present in the current manuscript version.
Author Response
Dear reviewer,
Thank you for considering our review article entitled Metabolomic profiling of CHO cells during the production of biotherapeutics for publication in Cells.
According to your comments:
- The Table 1 was transferred to Supplementary material. At this occasion it was converted to an Excel file to allow the reader to filter the studies for each parameter depending on one’s interest.
- As combining Tables 1, 2 and 3 resulted in a very heavy document that was complicated to read, we think that keeping them separated would facilitate the understanding. A “Comments” column was added to help the reader go through the differences that are observed for some amino acids.
- A graphical abstract was added. It includes the main hallmarks in metabolism, expected phase durations (days scale) and markers of phase change.
- Rephrasing was made at line 116 to clarify the point.
“This growth is possible as nutrients are available in the culture medium, from which catabolism generates energy and various substrates used for biomass generation.”
was changed to
“This growth is possible as nutrients are available in the culture medium, from which (1) catabolism generates energy and (2) various substrates are used for biomass generation.”
- Rephrasing was made at line 126 to clarify the point.
“Similarly, another study has concluded that glucose represents 65% of the entering carbon, along with 25% coming from amino acid and 10% from glutamine [43].”
was changed to
“Similarly, another study has concluded that glucose represents 65% of the entering carbon, along with 10% coming from glutamine and 25% coming from other amino acids and 10% from glutamine [43].“
- The mistake at line 323 was corrected. The sentence containing “Figure 2B” was deleted as it refered to a former version of the Figure.
Thank you
Round 2
Reviewer 1 Report
The conclusion was significantly improved according to the reviewers comments. It seems that there are some errors in the citations.